# Estimation of Leaf Area Index across Biomes and Growth Stages Combining Multiple Vegetation Indices

**DOI:** 10.3390/s24186106

**Published:** 2024-09-21

**Authors:** Fangyi Lv, Kaimin Sun, Wenzhuo Li, Shunxia Miao, Xiuqing Hu

**Affiliations:** 1State Key Laboratory of Information Engineering in Surveying, Mapping, and Remote Sensing, Wuhan University, Wuhan 430072, China; fangyilv@whu.edu.cn (F.L.); shunxiamiao@whu.edu.cn (S.M.); 2School of Information Science and Engineering, Wuhan University of Science and Technology, Wuhan 430081, China; liwenzhuo@wust.edu.cn; 3Key Laboratory of Radiometric Calibration and Validation for Environmental Satellites, China Meteorological Administration, Beijing 100081, China; huxq@cma.gov.cn

**Keywords:** leaf area index, vegetation index, MERSI-II

## Abstract

The leaf area index (LAI) is a key indicator of vegetation canopy structure and growth status, crucial for global ecological environment research. The Moderate Resolution Spectral Imager-II (MERSI-II) aboard Fengyun-3D (FY-3D) covers the globe twice daily, providing a reliable data source for large-scale and high-frequency LAI estimation. VI-based LAI estimation is effective, but species and growth status impacts on the sensitivity of the VI–LAI relationship are rarely considered, especially for MERSI-II. This study analyzed the VI–LAI relationship for eight biomes in China with contrasting leaf structures and canopy architectures. The LAI was estimated by adaptively combining multiple VIs and validated using MODIS, GLASS, and ground measurements. Results show that (1) species and growth stages significantly affect VI–LAI sensitivity. For example, the EVI is optimal for broadleaf crops in winter, while the RDVI is best for evergreen needleleaf forests in summer. (2) Combining vegetation indices can significantly optimize sensitivity. The accuracy of multi-VI-based LAI retrieval is notably higher than using a single VI for the entire year. (3) MERSI-II shows good spatial–temporal consistency with MODIS and GLASS and is more sensitive to vegetation growth fluctuation. Direct validation with ground-truth data also demonstrates that the uncertainty of retrievals is acceptable (R^2^ = 0.808, RMSE = 0.642).

## 1. Introduction

The leaf area index (LAI) is generally defined as the one-sided green leaf area per unit ground area in broadleaf canopies and the projected needle leaf area in coniferous canopies [1]. The LAI characterizes the density of leaves and vegetation canopy structure. It drives the microclimate in and under the canopy and controls canopy water interception, radiative extinction, and water and carbon gas exchange; it is therefore an important parameter in many climate, hydrological, biogeochemical, and ecosystem models [2,3].

There are many global LAI products, such as MODIS [4], GEOV1 [5], GIMMS 3 g [6], GLASS [7], GLOBCARBON [8], and VIIRS [9]. These products use reflectance data from satellites in the United States and Europe. In recent years, China’s polar-orbiting satellites for Earth observation have made significant advancements, providing new and valuable supplements to global environmental monitoring data. Among these, the Fengyun-3D (FY-3D) satellite, launched in November 2017, is a part of China’s second-generation polar-orbiting satellite series. Medium Resolution Spectral Imager-II (MERSI-II) is one of the main payloads of FY-3D [10], equipped with 25 bands ranging from visible to long-wave infrared, and can cover the Earth twice a day. MERSI-II is one of the most advanced low- and medium-resolution sensors today, and its high frequency of observation and wide coverage provide a new and reliable data source for global long-time-series LAI estimation.

Currently, LAI inversion methods based on optical sensors can be divided into three categories: empirical model method, physical model method, and machine learning method. The empirical model is simple and efficient but limited by season, terrain, vegetation type, and other factors, making it difficult to apply to the production of LAIs under large-scale and complex vegetation types. The physical model establishes the relationship between vegetation canopy reflectance and biophysical parameters at different wavelengths by simulating the photon radiation transmission process. The physical model has high universality but requires multiple parameters, and the inversion process is complex. In recent years, machine learning (ML) algorithms have gained popularity in LAI estimation due to their ability to efficiently handle large datasets and capture complex nonlinear relationships. Among the commonly used ML methods, backpropagation neural networks (BPNNs) [3,5,11,12,13,14,15,16], random forests (RFs) [12,14,17,18], and support vector machines (SVMs) [12,13,14] are the most widely adopted. Cao et al. [15] utilized a BPNN to model the relationship between GIMMS NDVI and Landsat LAI. Chen et al. [3] developed a BPNN-based LAI estimation model using an NDVI, biome map, longitude, and latitude as inputs. This model serves as a backup algorithm, activated when the radiative-transfer-based LAI inversion model fails. In another study, a BPNN was applied as a nonlinear regression method to relate reflectance, illumination geometry, and the LAI [5]. Tang et al. [12] employed an SVM, RF, and BPNN to construct estimation models for the winter wheat LAI and above-ground biomass using spectral vegetation indices as inputs.

Many experiments have shown that there is a strong correlation between LAIs and certain vegetation indices (VIs), so estimating LAIs using VIs has been widely used. However, different VIs have different characteristics. For example, the normalized difference vegetation index (NDVI) is susceptible to soil background at a low LAI and gradually saturates at a high LAI [19]; the soil-adjusted vegetation index (SAVI) and optimized soil-adjusted vegetation index (OSAVI) can reduce the effect of the soil background [20,21]; the enhanced vegetation index (EVI) is more sensitive in the region of high LAI values and weakens the effect of soil background and atmosphere [22]; the renormalized difference vegetation index (RDVI), modified simple ratio (MSR), and simple ratio index (RVI) have been proposed to mitigate the saturation effect in high-biomass regions [23,24,25]; and the difference vegetation index (DVI) is very sensitive to changes in soil background [26].

In recent years, there have been some studies exploring the relationship between different VIs and LAIs for specific vegetation types [27,28,29,30,31]. Dong et al. compared the potential of red-edge-reflectance-based (RE-based) and visible-reflectance-based (VIS-based) VIs to estimate the LAIs for spring wheat and canola [30]. They found that RE-based VIs were less sensitive to canopy structure compared to VIS-based VIs. Tillack et al. discovered seasonal variations in the VI–LAI relationship for trees [31]. Sun et al. analyzed the sensitivity of different VIs derived from multispectral instrument (MSI) and red-edge bands in estimating crop LAIs [32]. Viña et al. evaluated the estimation of LAIs for two crop types, maize and soybean, using various VIs [33]. However, most of these studies focused on one or two specific vegetation types and did not comprehensively consider the influence of vegetation type and growth stages on the VI–LAI relationship.

Our objectives are to (1) evaluate the variability of the VI–LAI relationship, particularly the VIs derived from the MERSI-II data; (2) estimate the LAI by self-adaptively combining multiple vegetation indices across various biomes; and (3) validate the retrieved MERSI-II LAI using MODIS and GLASS LAI products, as well as ground-truth data.

## 2. Materials and Methods

### 2.1. Study Area

The scope of the study area was concentrated in China, spanning from 73°33′ E to 135°05′ E and 17°30′ N to 53°33′ N, encompassing a wide range of latitudes and diverse vegetation types. The forest vegetation cover area accounts for 10.83%, and the non-forest vegetation cover area accounts for 62.78%. Detailed land cover classifications are presented in Figure 1, which includes a pie chart illustrating the proportional distribution of each land cover type.

### 2.2. Data Preparation

#### 2.2.1. MERSI-II Data Acquisition and Preprocessing

The MERSI-II data, obtained from the National Satellite Meteorological Centre’s official website (http://www.nsmc.org.cn/nsmc/cn/home/index.html (accessed on 1 January 2019)), is available from 2018 and beyond. MERSI-II has 25 bands, with 6 bands at a spatial resolution of 250 m and the remaining 19 bands at a spatial resolution of 1000 m. In this work, we only used blue, red, and near-infrared (NIR) bands, all with a spatial resolution of 250 m, to generate the VI.

The MERSI-II level 1 data were stored in HDF format, mainly providing earth observation solar reflection data, calibration coefficient, and geolocation fields. It is worth noting that we used the corresponding 1 km geolocation dataset because the geolocation dataset in 250 m resolution is not strictly corrected for terrain effects [34]. Then, the data need to go through the following pre-processing steps: (a) radiometric calibration; (b) atmospheric correction; (c) geometric reprojection (the MERSI-II image was re-projected to latitude/longitude coordinate system and resampled to a spatial resolution of 0.005°); (d) image stitching; and (e) cloud detection.

#### 2.2.2. MODIS Products and GLASS Products

In this study, MODIS LAI products and land cover maps were employed for the generation and validation of MERSI-II LAI, while GLASS LAI products served as an additional tool for cross-validation.

The latest MODIS LAI products (version 6) were released in 2015 and have been comprehensively evaluated and validated [35,36]. Considering that the FY-3D is an afternoon orbiting satellite and passes through at around 13:40 local time [10], the Aqua MODIS (afternoon overpass) MYD15A2H product was chosen for the estimation and validation of MERSI-II LAI.

MYD15A2H is an 8-day composite product, with the “best” LAI observations selected over 8 days, with a 500 m pixel size, containing 6 scientific datasets (SDSs). Among them, the first and second SDSs store the FPAR and LAI retrievals in turn; the third and fourth SDSs are quality control datasets indicating the algorithm paths (main and backup algorithms) and pollution information such as clouds, cloud shadows, snow, and aerosols. In this study, we used an LAI layer and two quality control layers. Cloud-free and low-aerosol LAI products retrieved from the main algorithm selected by quality control layers were used to analyze and model the VI–LAI relationship in this study. In addition, the MODIS LAI products were re-projected to a latitude/longitude coordinate system and resampled to a spatial resolution of 0.005°.

The MODIS Terra + Aqua MCD12Q1 product is an annual global 500 m sinusoidal grid product containing five different land cover classification schemes. Land Cover Type 3, which is a MODIS-derived LAI/FPAR scheme for the generation and validation of LAI products, was used in this study. It classifies global vegetation into eight major categories: grasses/cereal crops (Biome 1), shrubs (Biome 2), broadleaf crops (Biome 3), savanna (Biome 4), evergreen broadleaf forests (EBF, Biome 5), deciduous broadleaf forests (DBF, Biome 6), evergreen needleleaf forests (ENF, Biome 7), and deciduous needleleaf forests (DNF, Biome 8). The same re-projection operations were performed on the MCD12Q1 products.

The latest version of the GLASS LAI (V6) was released in 2022 [7]. It utilized a bidirectional long short-term memory (Bi-LSTM) recurrent neural network model to generate 23 years (2000–2022) of LAI data at 250 m and 500 m spatial resolutions. The GLASS LAI is currently the highest-spatial-resolution long-term global LAI product available.

#### 2.2.3. BELMANIP Network and Field Measurement Data

The cross-validation between MERSI-II and MODIS in this study was based on the BELMANIP (Benchmark Land Multisite Analysis and Inter-comparison of Products) ground validation network. The sites of the BELMANIP network are almost uniformly flat over a 10 × 10 km^2^ area and have a low ratio of urban areas and water bodies. They are highly stable in time series and are therefore widely used for cross-validation between multi-sensor data products [37].

This study employs the Ground-Based Observations for Validation (GBOV) LAI dataset as the ground reference to assess the accuracy of the estimated FY-3D MERSI-II LAI. For the LAI product, GBOV provides reference measurements (RMs) and corresponding upscaled land products (LPs), with information on 12 sites used in this study presented in Table 1. The LPs are derived from high-resolution satellite imagery by upscaling ground-based observations, with a spatial resolution of 300 m and coverage of 3 km × 3 km. This upscaling method has been utilized by the CEOS LPV group for the validation of global LAI satellite products.

In accordance with the quality control layers in the LP LAI dataset, pixels of poor quality were excluded to retain high-precision ground reference data for the validation of the retrieved FY-3D MERSI-II LAI. Only GBOV LP pixels corresponding to more than 70% of pixels in the high-resolution imagery with interpolation function as the transfer function for upscaling were utilized. Additionally, only GBOV pixels labeled as “within range” were considered, while those labeled as “out of range” were discarded.

### 2.3. Vegetation Index Extraction

Eight vegetation indices were directly computed from the preprocessed MERSI-II reflectance data to analyze the relationship with LAI (Table 2). The DVI, RVI, and NDVI are classical vegetation indices that estimate the LAI based on the difference and ratio of reflectance between the near-infrared and red bands. The RDVI, MSR, and EVI are improved vegetation indices derived by modifying the classical ones, which enhance the linearity with biophysical parameters and have a higher sensitivity to the LAI in high-biomass regions. The SAVI and OSAVI incorporate soil adjustment parameters to reduce the influence of soil background in areas with sparse vegetation.

The daily values of eight vegetation indices for two years, from 2018 to 2019, were calculated. Considering the seasonal differences in vegetation, the maximum value method was used to obtain 16-day composite vegetation indices. The maximum value composite method can partially eliminate the interference of clouds, shadows, aerosols, and other factors [38]. Furthermore, considering the random errors in MODIS LAI products, the average LAI was calculated using LAI products from 2010 to 2019, spanning a total of 10 years. The same method was applied to generate 16-day composite LAI products. Finally, 23 sets of corresponding 16-day composite vegetation indices and LAI products for one year were obtained.

### 2.4. Analysis of VI–LAI Relationship

In 2020, the four vegetation categories, namely grassland/cereal crops, savannah, deciduous broadleaf forest, and broadleaf crops, accounted for 40.40%, 18.09%, 5.50%, and 3.50% of China’s Mainland, respectively, making them the dominant biomes (Figure 1). The VI–LAI interannual time series curves for the eight VIs and four vegetation types were plotted separately to qualitatively analyze the correlation between different VIs and specific species’ LAI.

To further quantitatively analyze the VI–LAI relationship for specific species, two regression fitting methods, linear regression and logarithmic regression, were selected. The two fitting functions, including linear and curve models, help assess the linearity and non-linearity of the VI–LAI relationship for specific species. Considering the large span of latitudes and longitudes in the study area, as well as the significant differences in pixel counts for different vegetation types, representative and illustrative scatter plots were obtained using a random equal sampling method. A total of 800 pixels were randomly selected, with 100 pixels sampled from each of the eight vegetation types. The VIs and LAI values corresponding to the sampled pixels from 2018 to 2019 were collected to create density scatter plots and calculate the two fitting functions. Finally, the root mean square error (RMSE) and coefficient of determination (R^2^) were calculated as evaluation criteria for the fitting results.

### 2.5. Selection of the Machine Learning Models

#### 2.5.1. Data Preparation for Model Training

Quality control was applied to the MERSI-II vegetation index data, and only pixels with good atmospheric quality, no clouds, and low aerosol loading were selected. The MODIS LAI data from 2010 to 2019, spanning a total of 10 years, were averaged. Additionally, two quality control layers from the MYD15A2H product were used to select pixels retrieved by the main algorithm and not contaminated by clouds, shadows, and aerosols, so as to reduce the uncertainty in MODIS LAI products. In order to mitigate the interference of geographic registration errors between MERSI-II VIs and the MODIS LAI, the model was trained in a 15 × 15 window. Considering seasonal variations in vegetation, specific machine learning models were trained with a 16-day periodicity to estimate the daily LAI for corresponding dates. The processed 16-day composite MERSI-II VIs for 2018 and the 16-day composite MODIS LAI averaged over 10 years (2010–2019) were selected as training data. The study selected four representative dates in spring (DOY of 113–128 starting on 23 April 2018), summer (DOY 193–208 starting on 12 July 2018), autumn (DOY 289–304 starting on 16 October 2018), and winter (DOY 33–48 starting on 2 February 2018) of 2018 as the data sources for training and comparative analysis of three machine learning models.

#### 2.5.2. Model Training

BPNN [3,11,15], SF [12,14,17], and SVM [12,13,14] are widely used for estimating LAI based on remote sensing data. This study employs these three machine learning models to develop VI-based LAI estimation models.

(1) BPNN

The BPNN is an artificial neural network trained using the backpropagation algorithm [39]. The network consists of an input layer, one or more hidden layers, and an output layer. Neurons in each layer are connected through weights, which are adjusted by minimizing error. During training, the network first computes output values through forward propagation and then adjusts the weights and biases of each layer through backpropagation based on the error between the output and the actual values, thereby gradually reducing the error. In this study, a four-layer network was employed for training, consisting of an input layer (NDVI, biome type, latitude, and longitude), two hidden layers, and an output layer (LAI). The number of neurons in the two hidden layers was determined to be 10–15 and 5–10, respectively, through a grid-search-based hyperparameter tuning method. The activation functions for the hidden and output layers were ‘tansig’ and ‘purelin’, respectively. The Levenberg–Marquardt optimization algorithm was used as the training function.

(2) RF

Random Forest [40] is an ensemble learning method that enhances model accuracy and robustness by constructing multiple decision trees and combining their predictions. Each decision tree is trained on a random subset of the data and features, thereby mitigating the risk of overfitting. Ultimately, the random forest generates the final prediction by voting (for classification task) or averaging (for regression task) the predicted results of all decision trees. The number of decision trees in the random forest was optimized using a grid search approach.

(3) SVM

Support vector machine maps data into a high-dimensional feature space using kernel functions and utilizes hyperplanes to perform classification and regression tasks [41]. The penalty parameters C and γ were optimized using a grid search method. The kernel function was configured as ‘gaussian’.

### 2.6. LAI Estimation Based on Combined Vegetation Indices

We chose the model with the best performance to capture the nonlinear relationship between VI and LAI. The specific process is shown in Figure 2. First, the appropriate vegetation indices were selected. Among the eight vegetation indices mentioned, several MERSI-II vegetation indices suitable for large-scale and long-term LAI estimation were selected through time series analysis and regression fitting analysis. Second, considering that the sensitivity of the relationship between VI and LAI is influenced by species type and vegetation growth stage, we adopted a LAI estimation strategy that leverages combinations of optimal vegetation indices to maximize the sensitivity across different vegetation types and growth stages. Given the seasonal variability of vegetation, we considered 16 days as the growth phase and proceeded with training LAI estimation models in a phased manner. Multiple input parameters can be obtained by arbitrarily combining the selected vegetation indices. At each growth stage, species specific LAI estimation models were trained based on different input parameters. The best input parameters (i.e., a single vegetation index or a combination of multiple indices) were selected based on the R^2^ and RMSE values from the models’ test sets. This approach enables the identification of the most suitable vegetation indices for each of the eight vegetation types across their distinct growth stages. Embedding this as a screening criterion allows the model to select the optimal VIs for LAI estimation according to different vegetation types and growth stages.

The processed 16-day composite MERSI-II VIs for 2018 and the 16-day composite MODIS LAI averaged over 10 years (2010–2019) were used as training data. The machine learning model was trained to establish the relationship between the vegetation index and LAI. For each 16-day period in a year, a specific model for each species was trained. Once trained, the corresponding model could be used to estimate the MERSI-II LAI product by inputting the MERSI-II VI for each day.

## 3. Results

### 3.1. Relationship between the LAI and VIs

#### 3.1.1. Time-Series Analysis

The time-series changes in the NDVI, RVI, DVI, RDVI, MSR, EVI, SAVI, OSAVI, and LAI from 2018 to 2021 were plotted for four main biomes in China, as shown in Figure 3, Figure 4, Figure 5 and Figure 6. Taking Figure 3 as an example, the red line represents the LAI, and the blue line represents the vegetation index. The horizontal axis represents days 1 to 1461, covering the period from 2018 to 2021, while the vertical axis represents the average values of the vegetation index and LAI of all sampling points for the same biome.

Overall, the eight vegetation indices exhibit strong correlations with the LAI, displaying similar temporal variations and patterns to the LAI, effectively capturing the rapid increases, peaks, rapid declines, and trough phases of the LAI throughout the year. However, the correlations between various VIs and the LAI varied significantly across different biomes and growth stages. For grasses or cereal crops (Figure 3), the EVI effectively captures the peak of the LAI curve, while the response of other vegetation indices to the peak of the LAI exhibited a certain degree of lag. During the growth of broadleaf crops, the LAI experiences a brief plateau period (Figure 4). For this biome, the EVI demonstrates the highest consistency with the LAI and effectively captures the fluctuations during the plateau period, followed by the MSR. As seen in Figure 6, the peak points of the MSR and RVI are relatively consistent with those of the LAI.

Therefore, when estimating the LAI using vegetation indices, it is crucial to select the most sensitive indices based on the distinct characteristics of different vegetation types and growth stages, rather than generalizing.

#### 3.1.2. Regression Fitting Analysis

Linear and non-linear regression models were fitted to explore the relationship between the LAI and vegetation indices, as detailed in Table 3. It can be observed that various vegetation indices exhibit significant non-linear correlations with the LAI, as evidenced by higher coefficients of determination values in logarithmic regression, except for Biome 5. Among the indices, the EVI demonstrated the strongest correlation with the LAI, achieving the highest values in the regression models for each vegetation type, followed by the RDVI.

Taking grasses and cereal crops as an example, scatter plots of eight vegetation indices against the LAI were plotted (Figure 7). Vegetation indices tend to saturate in high-LAI regions, attributed to the reflectance saturation in dense vegetation areas [42]. When LAI values are low, the slope of the NDVI–LAI fitting curve is at its highest, indicating that the NDVI is most sensitive to LAI changes at lower biomass levels; as the LAI gradually increases, the NDVI saturates first. The scatter plots of EVI–LAI and RDVI–LAI exhibit more pronounced trends, denser clusters, and weaker saturation effects, indicating an ability to resolve LAI differences over a wider range of canopy conditions. The saturation effects of the MSR and RVI were weaker, with the MSR being more sensitive to LAI changes and showing a higher correlation with the LAI.

Based on a comprehensive analysis of qualitative temporal trends and quantitative regression fitting results, it was determined that four vegetation indices, namely the EVI, MSR, NDVI, and RDVI, were suitable for large-scale LAI estimation, considering the goodness of fit, sensitivity, and saturation effects.

### 3.2. Evaluation of Three Machine Learning Methods

Table 4 presents the performance of three machine learning models, BPNNs, random forests, and support vector machines, under four representative dates in 2018. For the four seasons of spring (DOY 113), summer (DOY 193), autumn (DOY 289), and winter (DOY 33), the R^2^ of the BPNN model on the test set were 0.7265, 0.7504, 0.8391 and 0.8105, respectively. In each season, the BPNN model consistently performed the best, with the highest R ^2^ and lowest RMSE. The accuracy of random forest is slightly inferior to the BPNN, while the SVM showed relatively lower performance. In addition, it can be seen from Table 4 that all three models demonstrated higher accuracy during the autumn and winter seasons, but lower accuracy during the spring and summer seasons. This can be attributed to the denser vegetation during spring and summer, with the NDVI being more sensitive when the LAI is low but gradually approaching saturation when the LAI is high.

Considering both R^2^ and RMSE, we chose to utilize BP neural networks to model the VI–LAI relationship for specific biomes and growth stages and estimate the MERSI-II LAI products.

### 3.3. Evaluation of the Estimated MERSI-II LAI

#### 3.3.1. Selection of Optimal VI Combinations for MERSI-II LAI Estimation

Based on the analysis in Section 3.1, it was found that the VI with the strongest correlation with the LAI varies across different biomes and growth stages. Therefore, when estimating the LAI using the VI–LAI empirical relationship, it is essential to select the optimal VI as an input parameter for specific species and growth stages.

The four vegetation indices, namely the NDVI, MSR, EVI, and RDVI, along with all their combinations listed in Table 5 were utilized as 15 distinct input parameters. For each growth stage (i.e., every 16 days) of each biome over one year, 15 BPNN models were trained, forming a set of estimation models. Using the MODIS LAI as reference data, the R^2^ and RMSE of different LAI results in each group were calculated (for combinations of two or three VIs, only the optimal was selected). Considering both R^2^ and RMSE comprehensively, the optimal VI for LAI estimation is shown in Figure 8. Therefore, we can adaptively select the optimal VIs for MERSI-II LAI estimation for different biomes and specific growth stages.

Figure 8 clearly shows that the optimal vegetation index for LAI estimation varies across different growth stages and biomes. For example, the combination of two vegetation indices is optimal for shrubs in winter; EVI works best for broadleaf crops in winter; for savannah, NDVI is ideal in autumn and winter, while EVI is better in summer; for evergreen needleleaf forest in summer, RDVI is most suitable.

#### 3.3.2. Comparison of LAI Estimation Using Combination of Multiple VIs and Single VI

To verify the superiority of the proposed method, LAI estimation models based on a single vegetation index, two VIs, three VIs, and four VIs were used to directly retrieve the LAI for the same year as the comparison methods. For two or three VIs, the one with the best results was selected, so there are seven comparison methods. The daily LAI of the study area in 2020 was retrieved using the proposed method and the comparison methods, respectively. For different biomes, the R^2^ and RMSE between the retrieved MERSI-II LAI and the corresponding MODIS LAI were calculated, and the annual mean values were calculated, as shown in Table 6 and Table 7. Five characteristic values of the R^2^ and RMSE for LAI estimation of all biomes throughout the year, namely the maximum, minimum, median, upper quartile, and lower quartile, were computed for each method, as shown in Figure 9.

For the eight biomes, the R^2^ of the proposed method is higher than that of the comparison methods, while the RMSE is lower, indicating that the accuracy and precision of the proposed method are better than the comparison methods (Table 6 and Table 7). Moreover, the estimation accuracy is the highest for Biome 1, Biome 2, and Biome 3, while it is lowest for Biome 4, Biome 5, and Biome 7. The result is consistent with the instability of LAI estimation caused by reflectance saturation in dense vegetation.

The differences between the estimated MERSI-II LAI and MODIS LAI were calculated, along with the mean absolute difference and standard deviation. The statistical results for four typical dates in 2020 (DOY of 081–096 starting on 21 March in spring, DOY of 193–208 starting on 11 July in summer, DOY of 289–304 starting on 15 October in autumn, and DOY of 017–032 starting on January 17th in winter) are listed in Table 8 and Table 9. The mean and STD of the LAI differences from the proposed method are significantly smaller than those of the comparison methods. This indicates that the retrieved MERSI-II LAI is close to the MODIS LAI, with fewer extreme differences, resulting in better estimation performance. Additionally, the mean and STD of LAI differences in summer are both high, indicating relatively poor estimation performance, which is consistent with the lush growth characteristics of vegetation in the Northern Hemisphere during summer.

### 3.4. Intercomparison between MERSI-II, MODIS, and GLASS

The MERSI-II LAI products for 2020 were produced using the proposed method, and the temporal and spatial consistency of MERSI, MODIS, and GLASS were evaluated.

#### 3.4.1. Spatial Consistency

The proposed method was employed to generate the 2020 MERSI-II LAI product, and the temporal and spatial consistency of MERSI, MODIS, and GLASS was evaluated. Figure 10a,b illustrate the mean and standard deviation of the differences between MERSI-II and MODIS (i.e., MERSI-II minus MODIS), while (c) and (d) depict these differences for MERSI-II and GLASS (i.e., MERSI-II minus GLASS). In these figures, gray pixels represent non-vegetation areas, and white pixels represent missing data due to cloud, shadow, and aerosol contamination. The results suggest that the mean and standard deviation of LAI differences exhibit similar spatial distribution patterns, with larger LAI differences corresponding to larger standard deviations. For both MODIS and GLASS, the ranking of the five categories of LAI differences is Category 3 > Category 2 > Category 4 > Category 1 > Category 5 (Table 10). Compared to MODIS, 83.27% of the retrieved MERSI-II LAI in the study area shows insignificant differences, indicating high consistency between the two products. Compared to GLASS, this proportion slightly decreases to 76.84%. Compared to the MODIS LAI, 11.82% of the MERSI-II LAI is underestimated; compared to the GLASS LAI, the underestimation rises to 22.02%. Since the MODIS LAI was used in training the retrieval model, this phenomenon is consistent with the findings of [43] that GLASS tends to overestimate the LAI in most forest areas. Qualitatively, the distribution of LAI differences aligns clearly with land cover (Figure 1). Specifically, for grassland and cereal crops, shrubs, and broadleaf crops, the absolute LAI differences between MERSI-II and both MODIS and GLASS are mostly less than 0.5, indicating good consistency. However, in forest areas, the MERSI-II LAI tends to be underestimated.

Figure 11 details the proportions of different LAI difference categories for each vegetation type. Figure 11a represents the results of MERSI-II minus MODIS, and Figure 11b represents the results of MERSI-II minus GLASS. The results for comparison between MERSI-II and MODIS as well as GLASS LAI products are relatively consistent. For grasses/cereal crops, shrubs, and broadleaf crops, the proportion of Category 3 is at least 73.62%, indicating significant agreement between the retrieved MERSI-II LAI and both the MODIS and GLASS LAIs. For savannas, the proportion of Category 3 remains the highest, but is below 70% compared to MODIS and drops to below 60% compared to GLASS, revealing higher uncertainty in LAI retrieval for savannas. For evergreen broadleaf forests, the proportion of Category 2 is the largest, nearing 50%, indicating that the MERSI-II LAI is underestimated in comparison to both the MODIS and GLASS LAIs. This may be due to reflectance saturation in dense canopies [42]. For deciduous broadleaf forests, the consistency between the MERSI-II LAI and MODIS LAI is relatively high. However, when compared to the GLASS LAI, the proportion of Category 2 reaches 55.02%. This is consistent with the findings of [43], which indicate that the GLASS LAI tends to be overestimated in deciduous broadleaf forests compared to MODIS. For evergreen and deciduous needleleaf forests, the MERSI-II LAI is also underestimated.

Figure 12 depicts the proportions of various vegetation types under each LAI difference category, offering an alternative quantitative analysis of the relationship between LAI differences and land cover types. Considering the uneven distribution of different biomes within the study area, proportional adjustments were applied during the analysis. The results indicate that evergreen broadleaf forests are the primary source of uncertainty in MERSI-II LAI retrieval. Compared to the MODIS LAI, their contribution proportions in Categories 1, 2, and 5 are 44.66%, 43.73%, and 79.47%, respectively. Compared to the GLASS LAI, their contribution proportions in Categories 1, 2, and 5 are 54.39%, 25.63%, and 51.74%, respectively. These findings are consistent with those in Figure 11. A comprehensive analysis of Figure 10, Figure 11 and Figure 12 suggests that vegetation type has a significant impact on the spatial distribution of LAI differences.

#### 3.4.2. Temporal Consistency

Figure 13 illustrates the temporal variations in the MERSI-II, MODIS, and GLASS LAIs for 2020 at BELMANIP sites for four different vegetation types: grasses/cereal crops, broadleaf crops, savannas, and deciduous broadleaf forests. The six sampling points, specific to different species, are well-distributed across mainland China. As evident from the figure, all three LAI products exhibit similar temporal variation trends at each site. Apart from site (a), the growth patterns of vegetation at the other sites follow typical Northern Hemisphere characteristics.

Among the three LAI products, GLASS, which uses bidirectional LSTM network prediction technology [7], demonstrates the best temporal completeness and smoothness. Nevertheless, the high smoothness also restricts its capacity to detect sudden anomalies. In contrast, MERSI-II and MODIS are affected by clouds, snow, and aerosol contamination, resulting in varying degrees of data gaps. For example, at site (a), MODIS lacks valid data, and MERSI-II exhibits gaps in winter data. Additionally, the MODIS LAI temporal curve exhibits large fluctuations, with many outliers inconsistent with vegetation growth patterns, as indicated by the red circles in Figure 13b–f. This is mainly due to the contamination of reflectance by clouds, aerosols, and other pollutants. In contrast, the reflectivity of MERSI-II is less affected by clouds, aerosols, and other factors. However, MERSI-II and MODIS are more sensitive in reflecting vegetation growth fluctuations. For example, at the blue arrow in Figure 13d, both MERSI-II and MODIS capture a trough, while GLASS shows a smooth curve. Overall, MERSI-II, MODIS, and GLASS exhibit good temporal consistency, indicating that the retrieved MERSI-II LAI has the potential for monitoring vegetation phenology changes.

### 3.5. Uncertainty Analysis for MERSI-II LAI Using Ground Measurements

Due to the lack of open-source ground-based measured data in the study area since 2018, we utilized GBOV sites in North America for direct validation of the estimated MERSI-II LAI. Figure 14 presents biome-specific scatter points and statistics (R^2^ and RMSE) of an FY-3D MERSI-II LAI against an GBOV LAI. The red dashed line represents the accuracy standard for LAI validation recommended by GCOS, indicating that when the ground reference LAI ≤ 2.5, the absolute difference between the validated LAI and reference LAI does not exceed ±0.5; when the ground reference LAI > 2.5, the relative difference does not exceed ±20%.

Overall, there is good consistency between the retrieved FY-3D MERSI-II LAI and ground reference LAI, with an R^2^ of 0.808 and RMSE of 0.642. Specifically, 75.2% of MERSI-II LAI data meet the accuracy requirements defined by GCOS. Spatial mismatch, gridding artifacts, ground measurement errors, scale uncertainties, auxiliary data errors, and other factors may affect the accuracy of LAI estimation [43]. Additionally, the distribution of scatter points indicates that LAI uncertainties are higher for evergreen broadleaf forests and deciduous needleleaf forests compared to grasses/cereal crops, shrubs, and savanna, consistent with cross-validation results at the site scale.

## 4. Discussion

Due to their simplicity and practicality, vegetation indices that combine visible and near-infrared reflectance bands have become widely adopted methods for estimating LAI. NDVI is one of the most classic and widely used vegetation indices. However, in moderate to high LAI regions, the strong absorption by chlorophyll leads to saturation of red-light reflectance, limiting NDVI’s sensitivity to green and greener vegetation. Compared to NDVI, EVI exhibits higher sensitivity to moderate and high LAI, but its performance is significantly affected by canopy structure complexity [28]. During the senescence stage of vegetation, the relationship between EVI and LAI tends to weaken [44]. Moreover, new vegetation indices like MSR and RDVI have alleviated the saturation problem of NDVI to some extent [23,24]. Notably, different vegetation indices show significant differences in sensitivity to the soil background, atmospheric effects, canopy structure, and leaf structure when estimating LAI. Extensive research on the VI–LAI relationship indicates that this relationship is species-specific and that vegetation indices exhibit varying sensitivity within the range of vegetation LAI [28,29,30,32,33]. Given the contrasting canopy architectures and leaf structures across different vegetation types and their growth stages, it is crucial to adopt an optimized vegetation index selection strategy based on growth stages and species characteristics to achieve higher accuracy in large-scale and long-term LAI estimations. As shown in Figure 9 and Table 5 and Table 6, selecting vegetation indices that best match specific vegetation stages and types can significantly improve the accuracy of LAI estimations.

The proposed method utilizes the MODIS LAI as reference data during the training phase; therefore, the accuracy and reliability of the MODIS LAI product become crucial factors affecting the model’s precision. Since it was released in August 2015, the Collection 6 (C6) version of the MODIS LAI product has undergone extensive validation. Compared to ground-measured LAI data, this product has a root mean square error of 0.66, and the deviation of most data points falls within the range of ±1 [36]. Cross-validation results with three major LAI products, namely GLASS, CYCLOPES, and GEOV1, indicate that MODIS is most consistent with GLASS [36,43]. Overall, the validation sites cover the major vegetation types worldwide and span a long period, suggesting that this product’s validation has reached the second stage [NASA]. Despite the high data quality of the C6 LAI product, it tends to underestimate high LAI values compared to measured LAI. Additionally, in densely vegetated areas and during the growing season in the Northern Hemisphere, the MODIS LAI shows larger differences compared to the other three products [36]. For broadleaf forest types, the MODIS LAI tends to overestimate at low values and underestimate at high values compared to the other three LAI products [36]. The inherent uncertainty of the MODIS LAI is one of the sources of error in the estimation model. Effectively integrating multiple LAI products to enhance the reliability of reference LAI is a key approach to further improving model accuracy. In addition, the study utilized the MODIS biome map as auxiliary data. However, land cover misclassification is one of the primary sources of uncertainty in LAI estimation models [2]. For instance, in MODIS LAI retrieval, misclassification in land cover can lead to incorrect algorithm selection, resulting in errors of 40–50%, while inaccuracies related to the representation of vegetation structure can cause errors of up to 20% [4]. Future research that integrates multiple land cover classification products to improve the accuracy of biome maps will further enhance the precision of LAI estimation models.

## 5. Conclusions

The Fengyun-3D satellite is one of the most advanced polar-orbiting satellites in China. The MERSI-II on board covers the world twice a day, providing a reliable data source for retrieving LAI for a wide range and high frequency. The VI–LAI relationship has been widely used in LAI estimation. However, most existing studies on the VI–LAI relationship focus on specific species and rarely consider the seasonal variation characteristics of vegetation, which is not suitable for large-scale and long-time-series LAI estimation.

In this study, we investigated the method of estimating MERSI-II LAI by combining multiple vegetation indices. Firstly, we evaluated the performance of three machine learning models—BPNN, RF, and SVM—in fitting the nonlinear relationship between the VI and LAI and found that BPNN performed the best. Next, we selected the suitable vegetation indices for eight vegetation types across different growth stages. For example, for LAI estimation of savannahs in autumn and winter, the NDVI is the optimal vegetation index. For evergreen broadleaf forests in spring, savannahs in summer, and broadleaf crops in winter, the EVI is the most suitable index for retrieving the LAI. For evergreen needleleaf forests in summer, the RDVI is the best index. The combination of two vegetation indices is more suitable for retrieving the LAI of shrubs in winter. Finally, by embedding the combination of the optimal vegetation indices as selection criteria into the BPNN model, adaptive estimation of MERSI-II LAI across biomes and growth stages can be achieved. Compared with the conventional method of using the same VIs for the estimation of the LAI for the whole year, the proposed method significantly improves the accuracy of the estimation results, with an R^2^ of 0.83 and an RMSE of 0.5 for all biomes in 2020.

Cross-validation with GLASS and MODIS indicates that the retrieved MERSI-II LAI demonstrates good spatio-temporal consistency with both products. Notably, MERSI-II LAI exhibits a more sensitive capability to capture vegetation growth dynamics, a characteristic that is particularly crucial for monitoring global environmental and climate changes. Furthermore, direct validation against ground measurements shows good agreement and acceptable uncertainty between MERSI-II LAI and ground-truth data. However, the inherent uncertainty of the MODIS LAI limits the model’s accuracy to some extent. Therefore, future research should focus on integrating multiple LAI products to further improve the accuracy and reliability of reference LAIs.

## Figures and Tables

**Figure 1 sensors-24-06106-f001:**
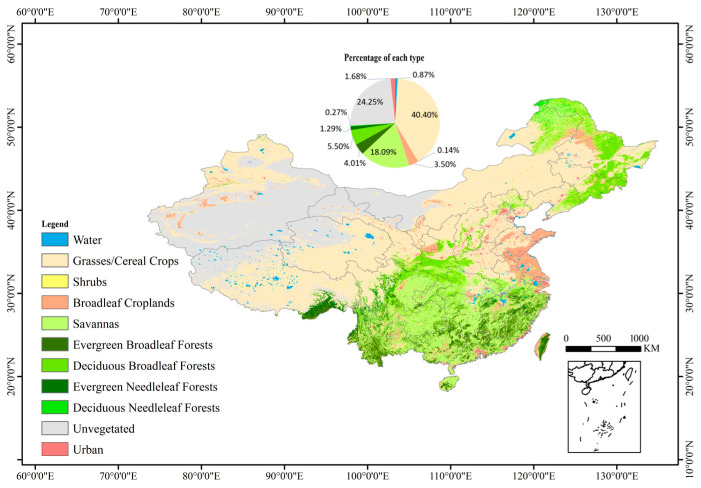
Land cover map of the study area using MODIS MCD12Q1 product in 2020.

**Figure 2 sensors-24-06106-f002:**
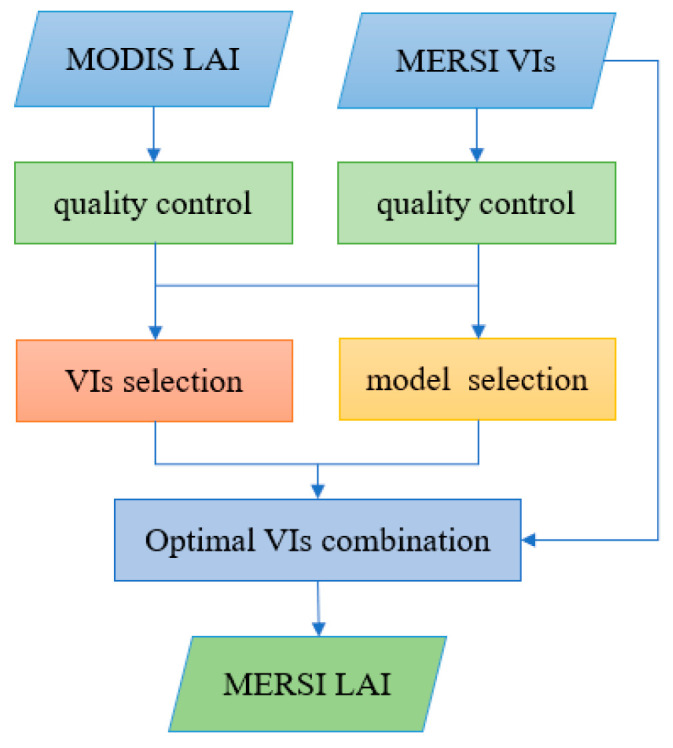
Flowchart for MERSI-II LAI estimation.

**Figure 3 sensors-24-06106-f003:**
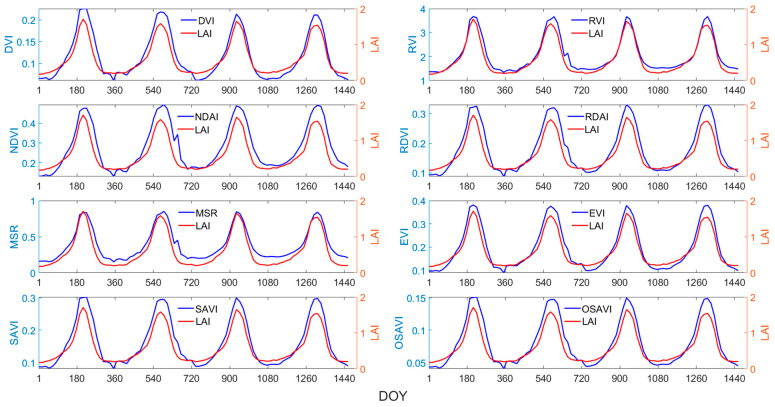
Time-series curves of VIs and LAI for grasses/cereal crops.

**Figure 4 sensors-24-06106-f004:**
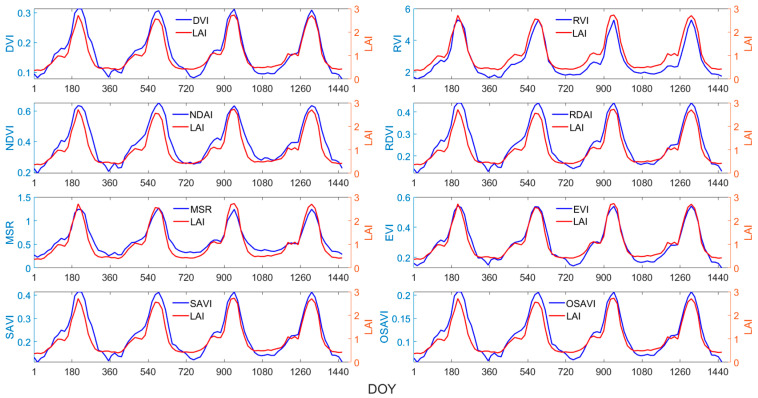
Time-series curves of VIs and LAI for broadleaf crops.

**Figure 5 sensors-24-06106-f005:**
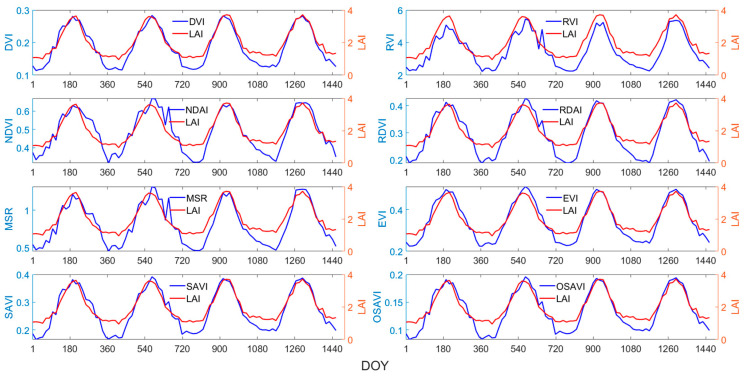
Time-series curves of VIs and LAI for savannahs.

**Figure 6 sensors-24-06106-f006:**
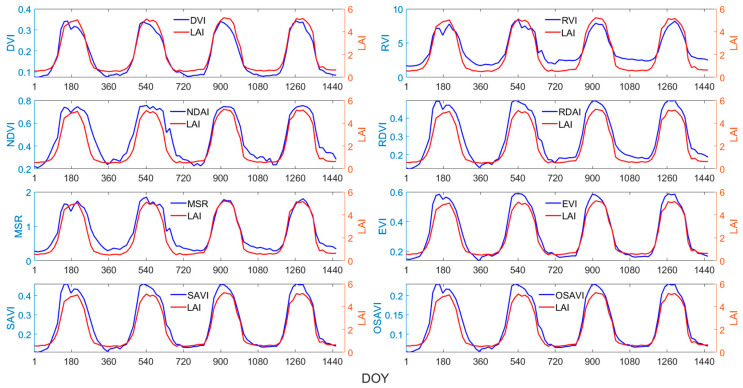
Time-series curves of VIs and LAI for deciduous broadleaf forests.

**Figure 7 sensors-24-06106-f007:**
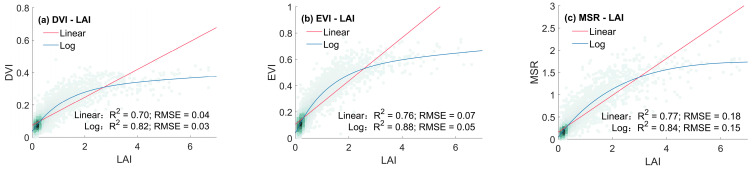
VI–LAI density scatter plot of grasses/cereal crops: (**a**) DVI–LAI; (**b**) EVI–LAI; (**c**) MSR-LAI; (**d**) NDVI–LAI; (**e**) OSAVI–LAI; (**f**) RDVI–LAI; (**g**) RVI–LAI; (**h**) SAVI–LAI.

**Figure 8 sensors-24-06106-f008:**
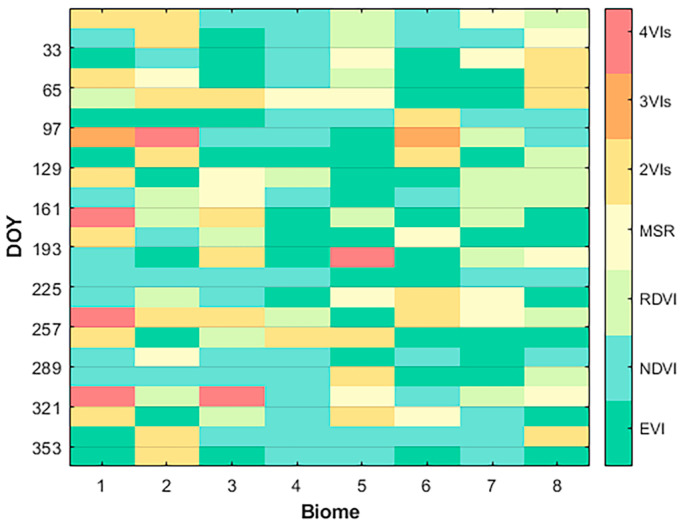
Optimal VIs for MERSI-II LAI estimation across different biomes and growth stages. Biomes 1–8 correspond to grasses/cereal crops, shrubs, broadleaf crops, savanna, EBF, DBF, ENF, and ENF, respectively. The seven colors represent seven different input parameters, respectively.

**Figure 9 sensors-24-06106-f009:**
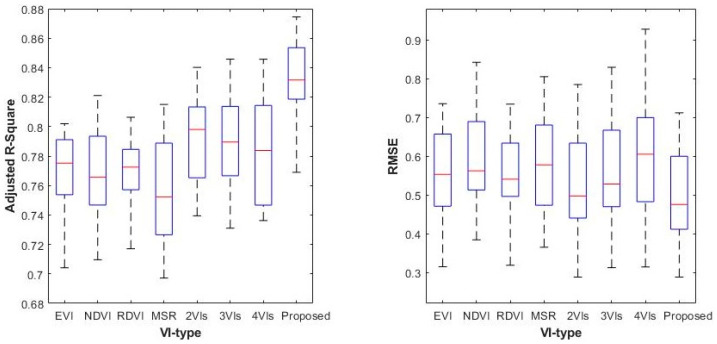
Estimation results of different methods.

**Figure 10 sensors-24-06106-f010:**
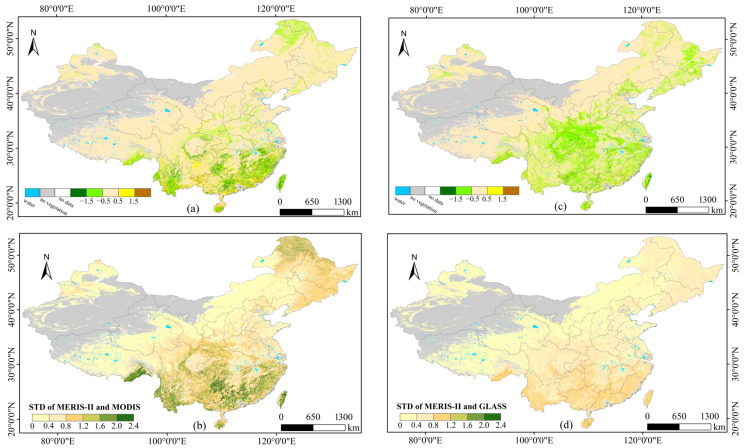
Comparison of spatial distributions of LAI differences between MERSI-II, Aqua MODIS and GLASS in mainland China in 2020: (**a**) MERSI-II LAI minus MODIS LAI; (**b**) STD of the LAI differences between MERSI-II and MODIS; (**c**) MERSI-II LAI minus GLASS LAI; (**d**) STD of the LAI differences between MERSI-II and GLASS. No-data pixels in white color are observations contaminated by cloud, shadow, aerosol, etc. Gray pixels are non-vegetation areas.

**Figure 11 sensors-24-06106-f011:**
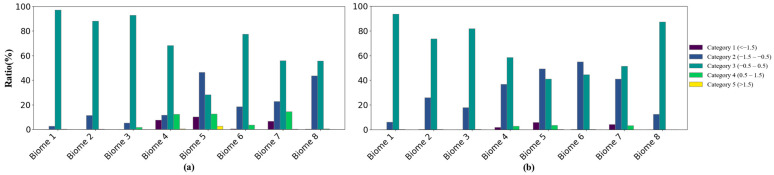
Bar chart for the proportion of different categories of LAI differences under each biome: (**a**) MERSI-II LAI minus MODIS LAI; (**b**) MERSI-II LAI minus GLASS LAI. Biomes 1–8 are grasses/cereal crops, shrubs, broadleaf crops, savanna, EBF, DBF, ENF, and ENF.

**Figure 12 sensors-24-06106-f012:**
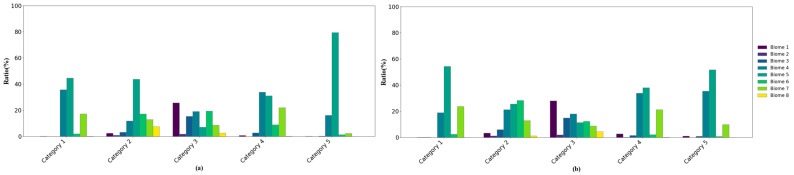
Bar chart for the proportion of different biomes under each category of LAI difference: (**a**) MERSI-II LAI minus MODIS LAI; (**b**) MERSI-II LAI minus GLASS LAI. Biomes 1–8 are grasses/cereal crops, shrubs, broadleaf crops, savanna, EBF, DBF, ENF and ENF.

**Figure 13 sensors-24-06106-f013:**
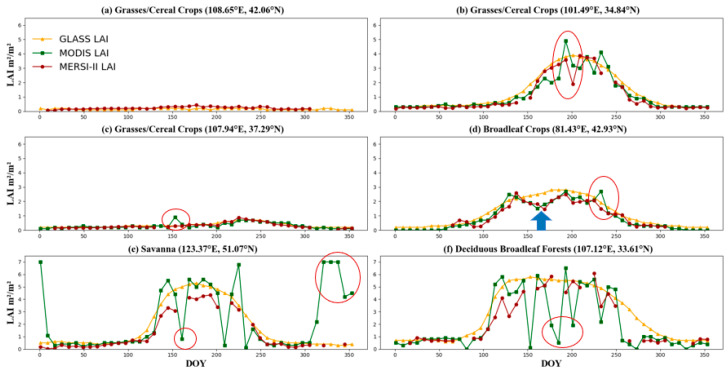
Time series of MERSI, GLASS and MODIS (2020). The red circle indicates large fluctuations in MODIS LAI.

**Figure 14 sensors-24-06106-f014:**
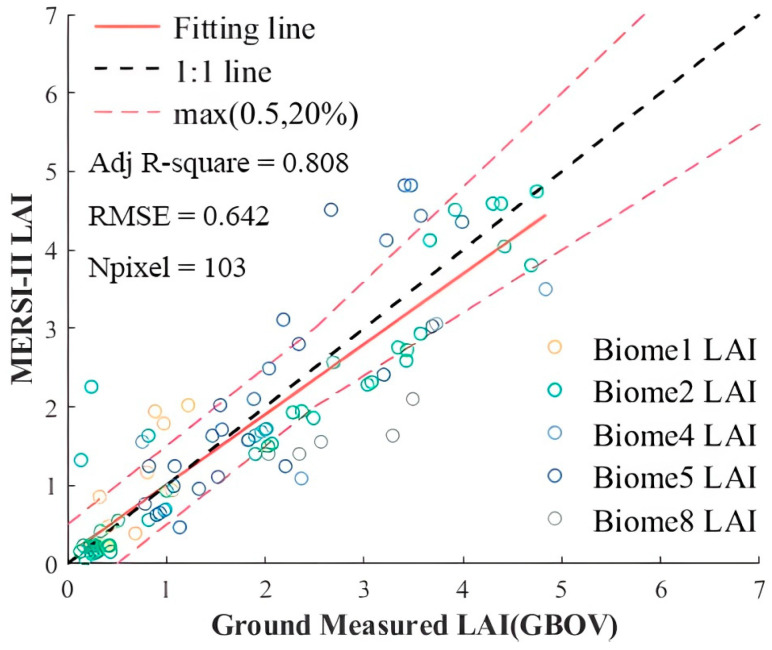
Comparison between ground-truth data and FY-3D MERSI-II LAI. Biome 1 is grasses/cereal crops, Biome 2 is shrubs, Biome 4 is savanna, Biome 5 is evergreen broadleaf forests, and Biome 8 is deciduous needleleaf forests.

**Table 1 sensors-24-06106-t001:** Information related to GBOV ground stations.

ID	Lat (°)	Lon (°)	Dominant Vegetation	Start Date–End Date
1	44.064	−71.287	savanna	03.18–10.12
2	39.060	−78.072	savanna	02.27–12.11
3	40.816	−104.746	grasses/cereal crops	03.12–10.06
4	32.542	−87.804	savanna	04.02–09.09
5	28.125	−81.436	evergreen broadleaf forests	01.26–11.25
6	17.970	−66.869	shrubs	01.13–12.14
7	31.195	−84.469	evergreen broadleaf forests	02.09–11.23
8	32.591	−106.843	shrubs	03.05–10.15
9	39.110	−96.613	evergreen needleleaf forests	03.02–10.12
10	18.025	−67.073	shrubs	01.13–12.30
11	40.177	−112.455	grasses/cereal crops	03.31–10.09
12	29.691	−81.997	evergreen broadleaf forests	01.01–11.16

**Table 2 sensors-24-06106-t002:** VIs for VI–LAI relationship analysis, where L = 0.5, $C_{1}$ = 6.0, and $C_{2}$ = 7.5 [20,22].

Index	Formula	Reference
RVI	RVI=Nir⁄Red	[25]
DVI	DVI=Nir−Red	[26]
NDVI	NDVI=(Nir−Red)/(Nir+Red)	[19]
RDVI	$RDVI=(Nir-Red)/\sqrt{Nir+Red}$	[24]
MSR	$MSR=(\frac{Nir}{Red}-1)/(\sqrt{\frac{Nir}{Red}}+1)$	[23]
EVI	$EVI=\frac{2.5\times (Nir-Red)}{Nir+C_{1}Red-C_{2}Blue+1}$	[22]
SAVI	$SAVI=(1+L)\times \frac{Nir-Red}{Nir+Red+L}$	[20]
OSAVI	OSAVI=(Nir−Red)/(Nir+Red+0.16)	[21]

**Table 3 sensors-24-06106-t003:** Regressions of the LAI on vegetation indices; bold indicates the highest correlation.

		DVI		EVI		MSR		NDVI		OSAVI		RDVI		RVI		SAVI	
		R^2^	RMSE	R^2^	RMSE	R^2^	RMSE	R^2^	RMSE	R^2^	RMSE	R^2^	RMSE	R^2^	RMSE	R^2^	RMSE
grasses/cereal crops	Lin	0.70	0.04	0.76	0.07	0.77	0.18	0.70	0.10	0.72	0.03	0.72	0.06	0.78	0.64	0.70	0.06
	Log	0.82	0.03	**0.88**	0.05	0.84	0.15	0.87	0.06	0.84	0.02	0.86	0.04	0.82	0.58	0.84	0.04
shrubs	Lin	0.77	0.03	0.80	0.06	0.81	0.18	0.75	0.10	0.78	0.02	0.78	0.05	0.83	0.61	0.78	0.05
	Log	0.87	0.03	**0.92**	0.04	**0.92**	0.12	**0.92**	0.06	0.91	0.02	**0.92**	0.03	0.90	0.47	0.91	0.03
broadleaf crops	Lin	0.67	0.05	0.70	0.09	0.73	0.22	0.67	0.10	0.69	0.03	0.67	0.07	0.72	0.83	0.68	0.07
	Log	0.81	0.04	**0.87**	0.06	0.81	0.18	0.85	0.07	0.85	0.02	0.85	0.05	0.75	0.79	0.84	0.05
savanna	Lin	0.49	0.06	0.54	0.09	0.50	0.31	0.46	0.12	0.49	0.04	0.51	0.08	0.49	1.28	0.50	0.07
	Log	0.63	0.05	**0.70**	0.07	0.60	0.27	0.65	0.09	0.65	0.03	0.68	0.06	0.56	1.19	0.66	0.06
EBF	Lin	0.30	0.06	**0.35**	0.08	0.16	0.36	0.14	0.10	0.29	0.04	0.29	0.07	0.17	1.63	0.32	0.07
	Log	0.30	0.06	**0.35**	0.08	0.17	0.36	0.15	0.10	0.29	0.04	0.29	0.07	0.17	1.63	0.32	0.07
DBF	Lin	0.78	0.05	0.81	0.08	0.75	0.33	0.68	0.12	0.78	0.03	0.77	0.07	0.71	1.54	0.77	0.07
	Log	0.83	0.05	**0.87**	0.07	0.81	0.29	0.82	0.09	0.85	0.03	0.86	0.06	0.75	1.45	0.84	0.06
ENF	Lin	0.38	0.05	0.45	0.08	0.23	0.34	0.22	0.12	0.37	0.04	0.36	0.07	0.27	1.40	0.36	0.07
	Log	0.39	0.05	**0.46**	0.08	0.25	0.34	0.27	0.12	0.40	0.04	0.39	0.07	0.28	1.39	0.38	0.07
DNF	Lin	0.76	0.04	0.79	0.06	0.78	0.26	0.70	0.13	0.76	0.03	0.77	0.06	0.78	1.00	0.78	0.05
	Log	0.84	0.03	0.86	0.05	0.86	0.21	0.86	0.09	0.86	0.02	**0.89**	0.04	0.82	0.91	0.87	0.04

**Table 4 sensors-24-06106-t004:** The accuracy of the test set for BPNN, RF, and SVM models.

DOY	BPNN		RF		SVM	
	R^2^	RMSE	R^2^	RMSE	R^2^	RMSE
113	0.7265	0.7834	0.7246	0.7889	0.6939	0.8768
193	0.7504	1.0775	0.7503	1.0780	0.7367	1.1366
289	0.8391	0.5933	0.8352	0.6077	0.8280	0.6341
33	0.8105	0.3605	0.8063	0.3685	0.7869	0.4054

**Table 5 sensors-24-06106-t005:** Eleven distinct combinations of four vegetation indices: EVI, NDVI, RDVI, and MSR.

Number	Combination of Vegetation Indices							
two VIs	EVI–NDVI	EVI–RDVI		EVI–MSR	NDVI–RDVI	NDVI–MSR		RDVI–MSR
three VIs	EVI–NDVI–RDVI		EVI–NDVI–MSR		EVI–RDVI–MSR		NDVI–RDVI–MSR	
four VIs	EVI–NDVI–RDVI–MSR							

**Table 6 sensors-24-06106-t006:** The annual mean R^2^ of LAI obtained using different estimation methods. And the bold indicates the highest R^2^.

	EVI	NDVI	RDVI	MSR	2VIs	3VIs	4VIs	Proposed
grasses/cereal crops	0.704	0.672	0.648	0.578	0.748	0.742	0.724	**0.810**
shrubs	0.606	0.657	0.575	0.584	0.737	0.722	0.712	**0.766**
broadleaf crops	0.627	0.614	0.594	0.547	0.691	0.663	0.646	**0.704**
savanna	0.437	0.465	0.439	0.417	0.506	0.493	0.487	**0.523**
EBF	0.330	0.265	0.316	0.268	0.325	0.307	0.295	**0.415**
DBF	0.476	0.426	0.451	0.390	0.478	0.440	0.437	**0.533**
ENF	0.445	0.398	0.420	0.350	0.490	0.443	0.407	**0.533**
DNF	0.249	0.166	0.231	0.226	0.336	0.189	0.255	**0.400**
All	0.767	0.765	0.764	0.728	0.783	0.789	0.767	**0.830**

**Table 7 sensors-24-06106-t007:** The annual mean RMSE of LAI obtained using different estimation methods. And the bold indicates the least RMSE.

	EVI	NDVI	RDVI	MSR	2VIs	3VIs	4VIs	Proposed
grasses/cereal crops	0.229	0.248	0.225	0.259	0.211	0.235	0.240	**0.193**
shrubs	0.162	0.232	0.181	0.241	0.161	0.185	0.196	**0.135**
broadleaf crops	0.341	0.357	0.327	0.385	0.298	0.346	0.345	**0.271**
savanna	0.629	0.719	0.645	0.679	0.625	0.707	0.729	**0.543**
EBF	0.632	0.768	0.678	0.671	0.680	0.790	0.764	**0.539**
DBF	0.335	0.429	0.339	0.439	0.337	0.379	0.400	**0.298**
ENF	0.576	0.685	0.615	0.658	0.529	0.653	0.640	**0.473**
DNF	0.239	0.320	0.258	0.259	0.173	0.226	0.206	**0.139**
All	0.559	0.590	0.551	0.616	0.552	0.558	0.591	**0.501**

**Table 8 sensors-24-06106-t008:** Mean absolute value of the difference between the MERSI-II LAI and MODIS LAI ($m^{2}/m^{2}$). And the bold indicates the minimum error.

DOY	EVI	NDVI	RDVI	MSR	2VIs	3VIs	4VIs	Proposed
017	0.268	0.310	0.283	0.317	0.286	0.282	0.281	**0.127**
081	0.251	0.310	0.278	0.294	0.254	0.297	0.271	**0.133**
193	0.540	0.539	0.525	0.574	0.530	0.534	0.548	**0.296**
289	0.549	0.601	0.548	0.549	0.540	0.547	0.552	**0.279**
mean	0.376	0.387	0.374	0.532	0.367	0.369	0.432	**0.182**

**Table 9 sensors-24-06106-t009:** STD of the difference between the MERSI-II LAI and MODIS LAI ($m^{2}/m^{2}$). And the bold indicates the minimum error.

DOY	EVI	NDVI	RDVI	MSR	2VIs	3VIs	4VIs	Proposed
017	0.509	0.546	0.513	0.547	0.521	0.507	0.507	**0.320**
081	0.485	0.525	0.505	0.550	0.480	0.500	0.505	**0.320**
193	0.806	0.824	0.808	0.822	0.793	0.791	0.800	**0.560**
289	0.879	0.886	0.874	0.860	0.850	0.841	0.846	**0.553**
mean	0.637	0.650	0.640	0.755	0.623	0.626	0.704	**0.402**

**Table 10 sensors-24-06106-t010:** The ratio of different LAI difference categories.

	Category 1 (<−1.5)	Category 2 (−1.5 – −0.5)	Category 3 (−0.5 – 0.5)	Category 4 (0.5 – 1.5)	Category 5 (>1.5)
MERSI-MODIS	2.64%	9.18%	83.27%	4.64%	0.27%
MERSI-GLASS	0.91%	21.11%	76.84%	1.11%	0.03%

## Data Availability

The datasets generated during and analyzed during the current study are available from the corresponding author upon reasonable request.

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
