# Peer review of "Estimation of Leaf Area Index across Biomes and Growth Stages Combining Multiple Vegetation Indices"

_sensors, 2024, doi:10.3390/s24186106_

Round 1
Reviewer 1 Report
Comments and Suggestions for Authors
1) L80: the authors claimed that growth stage was considered in this study. If this means the inclusion of LAI and VI time series in the training of BPNN, I do not think this method explicitly and comprehensively consider growth stages in its algorithm.
2) L85: in the method, there is no introduction of GLASS LAI.
3) L108: There may be something wrong with Fig. 1. According to the land cover map, most area in southeast China is covered by savannah, which is actually not true. For example, in Fujian province, most area should be classified as ENF or EBF. Moreover, what projection was used for the mapping?
4) VI-LAI relationships were established using linear, quadratic polynomial and logarithmic regression. However, these relationships were not used in retrieving LAI, but only BPNN were used for LAI estimation. So why were regressions performed?
5) What is the rationale of including coordinates in the BPNN?
6) Figs 3-6, “LVI” -> “LAI”. Are these figures drawn from the mean values of all pixels for each plant functional type?
7) Fig. 7, the quadratic regression should not be used for fitting VI-LAI relationship because the decline after the peak does not conform the reality, i.e. VI only saturates at high LAI but does not go down with LAI.
8) Figure and table captions are not detailed enough and hard to read. For example, Table 3: what are these B1 to B8. I know it may be explained in the text, however it makes the reading process much easier if the explanation also appears in the caption.
9) L212, VIs are in the year of 2018 and MODIS LAI is the average over 2010-2019? Considering the interannual variations in LAI and corresponding VIs, I do not think this is correct.
10) L274-275: Now I realize that the regressions of VI-LAI were performed with the aim of optimal VI selection. I have two questions here: 1) based on what criteria were these four VIs selected for LAI inversion using BPNN? 2) if the BPNN were established for each land cover type, then why not select different VI combinations for different species? For example, in Table 3, for B5, VIs with highest R-square are DVI, EVI, RDVI and SAVI, why do not select them to train BPNN rather than the four VIs selected in the text?
11) Table 4 and Line 284 are even more confusing. How were two-VI combination used as one input parameters?
12) L305: so the estimated MERSI-II LAI is in 2019? However, in L338, the estimated MERSI-II LAI is for 2020.
13) Figure 9, I guess the proposed method means “using 15 distinct input parameters”? If so, no wonder the accuracy of the “proposed” method is much higher than other methods, since the “proposed” method used more VIs combinations.
14) L324: if I read it correctly, the BPNN were established based on VIs in 2018 and average LAI between 2010-2019. Now the validation of estimated LAI is against MODIS LAI in 2019. I do not think this is the right way to prove the accuracy of the algorithm. The right way is to use one year of data for algorithm development and the data from a different year for validation of results.
15) Tables 10 and 11 can be expressed as bar charts, making the numbers much easier to understand.
16) L416: Fig. 4d
17) L415-417: Without in-situ LAI measurements, how can we know the decrease between Day 150 and 200 is reality or just uncertainties in the RS LAI.
18) In Fig. 11, MERSI-II LAI does not exhibit sudden drops as in MODIS LAI, the reason should be analyzed. It is probably due to the reflectance data of MERSI-II is less affected by factors such as aerosol and cloud compared to MODIS.
19) Fig. 12, what are the validation results for MODIS and GLASS LAI? Instead of Biome1-8, please specify land cover type in all figures.
20) L456-457: as stated in my last comment, the higher accuracy of the “proposed” method than other methods is because, the proposed method includes more VIs combinations and hence more information, not because of the usage of optimal VIs.
21) L484-495:L The drawn conclusions here cannot be supported by the results in Section 3.2.2
22) Most importantly, it was claimed in the beginning of this manuscript that “most of these studies… did not comprehensively consider the influence of vegetation type and growth stages…” as in L79-81. Then, the “vegetation type” and “growth stages” should be the possible of novelty of this manuscript, and therefore most of the discussion and conclusions should be focused on this aspect. However, I did not see much discussion on this point in this manuscript.
Comments on the Quality of English Language
The English language of this manuscript is good, the main probably of the manuscript is the content and logic flow.
Reviewer 2 Report
Comments and Suggestions for Authors
This manuscript presents a sufficient work which aims to retrieve LAI products from MERSI-â…¡, and it considers the influence of different vegetation types and growth stages on the VI-LAI relationship. The results are mainly compared with MODIS, and GLASS LAI, demonstrating good temporal consistency. In general, it is well written and organized, while some details are missing. Detailed comments are as follows.
Page 2, line 68, should RVI be SRI? please recheck the terms and language to remove the other potential errors.
Section 2.2.2, line 113, The latest MODIS LAI products (version 6) were released in 2015
Section 2.5 line 205, The MODIS LAI data from 2010 to 2019, spanning a total of 10 years
It is unreasonable that the latest MODIS LAI products appeared in 2015, but the training data covered 2015-2019, I wonder if it is the different MODIS LAI data version, or this is a mistake.
Section 2.5, suggest explain why the MERSI-II VIs for 2018 and the MODIS LAI data from 2010 to 2019 were chosen as the training set.
Section 2.2.1 and Section 2.5, the resampled spatial resolutions for both MODIS LAI and MERSI-II VIs are 0.005°, is that optimal for the training data?
Section 2.5, please cite more latest references on the application of BPNN and other machine learning methods in remote sensing. The reverse neural network employed did not reveal any mechanism and implementation of the backpropagation principle, and merely encompassed a traditional neural network model.
Section 3.1,the results show the time-series variation of the Vegetation Index (VI) for only two years, 2018-2019. It is recommended that the analysis be expanded to include additional years, such as 2019-2021, or be extended to a longer time period.
Figure 8, the targeted cells represent optimal VI or VIs. To ascertain the component of optimal VIs, please supplement the comparison of R² and RMSE of two and three detailed VIs
Figure 11(e,f), the broken line represented by MODIS exhibits extremely intense fluctuations. Compared with MERSI and GLASS, the disparity is significant. Then, can it be claimed to have good temporal consistency?
Reviewer 3 Report
Comments and Suggestions for Authors
The manuscript entitled “Estimation of Leaf Area Index across Biomes and Growth Stages Combining Multiple Vegetation Indices” (manuscript number “sensors-3156005”) was reviewed carefully. The manuscript investigates the LAI estimation based on vegetation indices derived from MERSI-II satellite images and compares its performance with the MODIS LAI product. The research uses an artificial neural network (multi-layer perceptron) to estimate LAI for eight biomes in China from 2018 to 2019.
Utilizing data from a recently (compared to MODIS) launched satellite mission to compute the LAI, which serves as an alternative for the MODIS LAI product, along with the employed methodology, is highly advantageous. However, I have serious concerns about the current manuscript version. Before publication, it underwent improvements to benefit the journal's readers. I express my concern in the manner described below:
- The introduction section lacks sufficient literature on LAI estimation using VIs and deep learning methods.
- MODIS LAI data was used from 2010 to 2019 and averaged, while MERSI-II data was used only from 2018 to 2019. What is the rationale for using 10 years of data from the MODIS product and comparing one year of LAI estimation values from MERSI-II?
- How many neurons are used in both hidden layers? Which activation function is used? How do you decide these parameters? Have the authors used any hyperparameter tuning methodologies, such as grid search or Bayesian hyperparameter optimization?
- The authors used only one neural network algorithm and compared the results with standard regression methods. What about the results of other classical machine learning algorithms like random forest, support vector machine, etc.?
- The accuracy metrics of standard regression methods and neural networks are close to each other. Why should one use complex neural network methods rather than simple regression methods?
Additionally, I would like to share my brief observations on:
- I would suggest using the term “estimation” rather than inversion. Because the study's main objective is to estimate LAI with derived VIs, it is a data-driven method, not an inversion.
- Bands or channels? Please use just one of them. I would suggest the term band.
- The time series figures should go to the materials and methods section, with a subsection called data explanatory analysis. These figures solely assess the time series of covariates and the target variable.
Comments on the Quality of English Language
The manuscript has an understanding of the English language.
Round 2
Reviewer 1 Report
Comments and Suggestions for Authors
Although the authors made efforts to improve the quality of the manuscript, however, some of my concerns still have not been resolved, and I do think the authors can resolve in the current manuscript.
One among these concerns would be the use of a flawed land cover map as in Fig. 1.
Another major concern is the 'claimed' novelty which is the inclusion of vegetation types in the algorithm. Actually, most of the current LAI products is based species-specific algorithms, then the consideration of vegetation types is not a new story. The use of season-specific model could be something new, although its better performance is still predictable.
The clarity of the writing could be improved in many ways.
Comments on the Quality of English Language
The language is ok for this manuscript, however, the clarity could be improved.
Reviewer 3 Report
Comments and Suggestions for Authors
The manuscript is significantly improved compared to the first version. I only have minor recommendations,
P.3 L.119-120: “MODIS LAI products and land cover map were used in this study for MERSI-â…¡ LAI generation and validation. And GLASS LAI products are used for cross-validation.” Please rewrite this sentence and merge them for clear readability.
Please add relevant citations for BPNN, RF and SVM methodologies in the section “2.5.2. Model Training”.
Comments on the Quality of English Language-
